# Screening of Entomopathogenic Fungal Culture Extracts with Honeybee Nosemosis Inhibitory Activity

**DOI:** 10.3390/insects14060538

**Published:** 2023-06-09

**Authors:** Dong-Jun Kim, Ra-Mi Woo, Kyu-Seek Kim, Soo-Dong Woo

**Affiliations:** 1Department of Agricultural Biology, College of Agriculture, Life & Environment Science, Chungbuk National University, Cheongju 28644, Republic of Korea; djkim@prolagen.com (D.-J.K.); loverami92@naver.com (R.-M.W.); zpdlems@naver.com (K.-S.K.); 2Process Development Team, R&D Division, Plolagen Co., Ltd., Seoul 03722, Republic of Korea

**Keywords:** honeybee, nosemosis, *Nosema ceranae*, *Apis mellifera*, entomopathogenic fungi, culture extracts

## Abstract

**Simple Summary:**

Nosemosis in honeybees caused by the *Nosema* spp. infection is not only related to colony collapse but also increases the susceptibility of honeybees to various other pathogens; thus, it is necessary to develop effective control methods for nosemosis. Research on various control methods has been conducted, but clearly defined methods are still rare. In our study, the possibility of controlling honeybee nosemosis using entomopathogenic fungal culture extracts with various biological activities was evaluated. Many of the tested entomopathogenic fungal culture extracts showed high germination inhibitory activity on *Nosema ceranae* spores. Among them, fungal culture extracts with high spore germination inhibitory activity and fungicidal activity was used to evaluate honeybee nosemosis inhibitory activity. As a result, it was confirmed for the first time that the fungal culture extract could not only suppress honeybee nosemosis but also increase the lifespan of honeybees. Entomopathogenic fungal culture extracts are expected to have applications in controlling honeybee nosemosis.

**Abstract:**

This study aimed to select the most effective culture extracts for controlling honeybee nosemosis using 342 entomopathogenic fungi of 24 species from 18 genera. The germination inhibitory activity of the fungal culture extract on *Nosema ceranae* spores was evaluated using an in vitro germination assay method. Among 89 fungal culture extracts showing germination inhibitory activity of approximately 80% or more, 44 fungal culture extracts that maintained their inhibitory activity even at a concentration of 1% were selected. Finally, the honeybee nosemosis inhibitory activity was evaluated using the cultured extracts of five fungal isolates having a *Nosema* inhibitory activity of approximately 60% or more, even when the extract was removed after treatment. As a result, the proliferation of *Nosema* spores was reduced by all fungal culture extract treatments. However, only the treatment of the culture extracts from *Paecilomyces marquandii* 364 and *Pochonia bulbillosa* 60 showed a reduction in honeybee mortality due to nosemosis. In particular, the extracts of these two fungal isolates also increased the survival of honeybees.

## 1. Introduction

The honeybee (*Apis mellifera* L. (Hymenoptera: Apidae)) performs an important role in the pollination of flowering plants, which are essential for the production of fruits, nuts, and seeds upon which animals, including humans, rely for food [1,2,3,4]. In addition to their role in agriculture, honeybees are also important for maintaining biodiversity in natural ecosystems [5]. They help pollinate wildflowers, which provide habitat and food for a variety of other species. Honeybees are also vital to the production of honey, beeswax, and other bee products that humans have used for thousands of years [6,7]. However, unfortunately, honeybee colony collapse has become frequent, with global bee populations rapidly declining in recent years due to habitat loss, pesticide use, climate change, and disease [8,9]. This is a major concern for the health of both food systems and natural ecosystems. Efforts are therefore underway worldwide to protect and conserve honeybee populations, including promoting habitat restoration, reducing pesticide use, and supporting research on bee health and behavior [9].

Bee colony collapse is a complex phenomenon that can be caused by multiple environmental and human-related factors [8,9,10]. Habitat loss and fragmentation, climate change, pesticides, mites, and other parasites, and diseases and pathogens have been highlighted as major causes of honeybee colony collapse [11,12,13,14]. Notably, bee colony collapse is often the result of a combination of these factors rather than a single cause. To address honeybee colony collapse, it is important to take a multipronged approach that addresses both environmental and human-related factors. Of these factors, honeybee disease, known as nosemosis, is known to have the greatest impact on honeybees [11,14,15,16]. *Nosema* infection can have severe negative effects on honeybee colonies, including reduced foraging activity, decreased colony growth and productivity, increased mortality, increased susceptibility to other stressors, reduced winter survival, decreased queen bee productivity, and reduced immune function [16,17,18,19,20,21,22,23]. Nosemosis is caused by two species of microsporidia, *Nosema apis* and *N. ceranae*. *Nosema ceranae* is more common and more lethal than *N. apis* [16,24,25,26]. Between the prevention and control of nosemosis, preventive methods are mainly relied on. Representative preventive methods include good beekeeping practices, such as keeping the hive clean and providing adequate nutrition and moisture to the bees, hygienic behaviors, such as removing diseased and dead brood from the hive, and avoiding stressors, such as exposure to pesticides and poor nutrition [9,27,28,29]. In contrast, for direct control, an antibiotic called fumagillin is used, but its use is controversial due to its potential negative effects on the environment and other organisms [30,31,32,33]. Research efforts to replace it have reported that probiotics, plant essential oils, propolis, plant extracts, royal jelly, etc., can reduce the production of *Nosema* spores in honeybees and improve the survival rate of *Nosema*-infected bees [22,23,34,35,36,37].

Entomopathogenic fungi, which are fungi that infect and kill insects, have been widely studied and used as materials for microbial insecticides for pest control [38,39,40,41]. Metabolites of these entomopathogenic fungi have various physiological activities, such as insecticidal activity, antibacterial activity, antioxidant activity, immunomodulatory activity, and cytotoxic activity, and potential applications are being sought in various fields, such as agriculture, medicine, and biotechnology [42,43,44,45].

In our study, entomopathogenic fungal culture extracts with proliferation inhibitory activity against *N. ceranae* were screened from various entomopathogenic fungi for the prevention and control of nosemosis, and the effect of improving the survival rate of honeybees upon *Nosema* infection was evaluated. The purpose of this study was to provide basic data on the possibility of using culture extracts of entomopathogenic fungi for the prevention or control of nosemosis in honeybees.

## 2. Materials and Methods

### 2.1. Honeybees and N. ceranae

A colony of the honeybee *A. mellifera* was reared with 50% (*w*/*v*) sucrose solution, and bee bread as the main food, and adults that emerged within 24 h were used in the experiment. Honeybees highly infected with *N*. *ceranae* were provided by the Sericulture and Apiculture Division of the Rural Development Administration, Republic of Korea.

### 2.2. Entomopathogenic Fungal Culture Extract

In this study, 342 isolates of entomopathogenic fungi from 18 different genera were used (Appendix A) [46]. In the same fungal species, fungal isolates indicate particular fungi from a particular environment or region. Fungal isolates were initially suspended in 1 mL of Sabouraud dextrose broth medium containing yeast extract (SDYB: 10 g of Bacto peptone, 40 g of dextrose, 10 g of yeast extract in 1000 mL of distilled water, and pH 6.0). Cultures were inoculated with agar blocks (6 mm in diameter) of fungi cultured in potato dextrose agar (PDA) medium for 2 weeks and grown in the dark at 25 °C with shaking at 150 rpm. After 10 days, the cultures were centrifuged at 10,000× *g* for 10 min. After removing the pellet, the supernatant was filtered using a LaboPass™ Mini Plasmid DNA Purification Kit column (Cosmo Genetech Co. Ltd., Seoul, Republic of Korea) to remove spores and mycelia. The ethyl acetate fractionation method was used to separate hydrophobic substances from fungal culture filtrates. After adding the same volume of ethyl acetate as the culture medium, vortexing for 20 min, and centrifugation at 4000× *g* rpm for 5 min, the supernatant was collected. Afterward, ethyl acetate was evaporated using gaseous nitrogen, and the remaining extract pellet was dissolved in 2% acetone in the same volume as the culture filtrate and used in the next experiment. The prepared culture extract was stored at −76 °C until use. To prepare a culture extract after quantitative inoculation of entomopathogenic fungi, fungal conidia that were harvested after being cultured in PDA medium for 2 weeks were used to prepare a conidial suspension using a 0.02% Tween-80 solution. The conidia were then counted using a hemocytometer. The conidial suspension was inoculated in 30 mL PDB medium in 50 µL at a concentration of 9 × 10^5^ conidia/mL and cultured for 10 days. After culturing, spores and mycelia were removed from the culture medium, and culture extracts were prepared using the ethyl acetate fraction method as described above.

### 2.3. Purification of Nosema Spores

To produce infective spores, honeybees were placed in plastic cages and inoculated with 1 × 10^6^ spores of *N. ceranae* in sucrose solution (50% *w*/*v* in water). To obtain purified *Nosema* spores, after 10 days, the midgut tissues from heavily infected honeybees were individually separated using forceps and washed with phosphate-buffered saline [47]. The isolated midgut was ground in 200 μL of sterile distilled water in a Bullet Blender^®^ Homogenizer (Scientific Instrument Services Inc., Palmer, MA, USA) set to speed 8 with 2 mm diameter tungsten carbide beads (Sigma–Aldrich, St. Louis, MO, USA) for 2 min. The homogenate volume was increased to 1 mL to increase the filtration efficiency, and the mixture was filtered through Qualitative No. 2 filter paper (Advantec MFS Inc., Dublin, OH, USA) with an 8–11 μm pore size to remove tissue debris [48]. The filtered suspension was overlaid very gently on discontinuous 25%, 50%, 75%, and 90% Percoll^®^ (Sigma–Aldrich, St. Louis, MO, USA) gradient and centrifuged twice at 15,000× *g* for 30 min at 20 °C [49]. A small but dense band just above the bottom of the tube was collected and resuspended in sterile distilled water. After final centrifugation at 15,000× *g* for 10 min at 20 °C, the spore pellet was resuspended in sterile distilled water. The spore concentration was measured by counting with a hemocytometer [50]. The viability of *Nosema* spores was determined by the in vitro germination method as described below, and those with a viability of 95% or more were used in the experiment.

### 2.4. In Vitro Germination Assay

Aliquots of purified *Nosema* spores (10 μL; 1 × 10^3^ spores) were spotted onto glass slide reaction cells (12 wells; Paul Marienfeld GmbH & Co. KG, Lauda-Königshofen, Germany) and air-dried for 2 h at room temperature. Germination was triggered by adding 1.5 μL of 0.1 M sucrose in distilled water to the air-dried spores [51]. After maintaining the covered glass slide at room temperature for 6 h, the germinated spores were observed under a light microscope (magnification, 400×) (Nikon Instech Co., Ltd., Tokyo, Japan). The germination rate was calculated as the percentage of total observed spores that had germinated.

### 2.5. Safety Test of Fungal Culture Extract

Twenty honeybees that emerged within 24 h were transferred to one cage and allowed to adapt for 3 days before being used in the experiment. To evaluate the safety of the fungal culture extract, 50% (*w*/*v*) sucrose solution mixed with culture extract was fed once to the honeybees, and as a control for comparison, 50% (*w*/*v*) sucrose solution containing the same concentration of acetone used to dissolve the culture extract was fed once. The survival rate of honeybees was observed and recorded every day for 14 days after treatment, and the experiment was repeated three times for each treatment group.

### 2.6. Inhibitory Activity of Fungal Culture Extracts on Nosema Infection in Honeybees

To evaluate the effect of fungal culture extract on *Nosema* infection in honeybees, 25 honeybees that emerged within 24 h were used for each experimental group. Honeybees were orally infected with purified *Nosema* spores by making a 50% sucrose suspension at 1 × 10^6^ spores/mL and administering 1 mL to each experimental group. Treatment of fungal culture extract was carried out before and after *Nosema* inoculation. For the control group, honeybees were infected with *Nosema*, and no culture extract treatment was used. To evaluate the inhibitory effect of *Nosema* infection, the survival rate of honeybees was observed and recorded every day for 14 days. The production of *Nosema* spores was examined by isolating and counting spores from the midgut of honeybees on the 14th day after infection to evaluate the effect of inhibition on *Nosema* growth. The experiment was repeated three times for each treatment group.

### 2.7. Statistical Analysis

Spore germination and honeybee survival rate results were analyzed with SPSS statistical software v12.0 (SPSS, Inc., Chicago, IL, USA). Data were subjected to a one-way analysis of variance (ANOVA), and comparisons among groups were performed with the SNK test. Data are expressed as the means ± standard errors (SEs), and statistical significance was set at the conventional α < 0.05 level.

## 3. Results

### 3.1. Inhibitory Activity of Entomopathogenic Fungal Culture Extracts on Nosema Germination

To evaluate the inhibitory activity of the entomopathogenic fungal culture extract on *Nosema* spore germination, the influence of acetone, a solvent used in preparing the culture extract, on *Nosema* spore germination was first evaluated. As a result of treating *Nosema* spores with various concentrations of acetone from 0.25% to 20%, no significant influence of acetone on the germination of *Nosema* spores was observed at all concentrations (Appendix A). Therefore, it was confirmed that the 0.2% concentration of acetone used in our experiment had no influence on the germination of *Nosema* spores. All fungal isolates used in the experiment are shown in Appendix A. As a result of evaluating the germination inhibitory activity of the culture extracts of 342 fungal isolates on *Nosema* spores, the germination of spores was inhibited in a variety of ways from 0% to a maximum of approximately 96.5% (Appendix A). Most of the spores did not germinate when the germination inhibitory activity was greater than approximately 80%, some spores germinated when the germination inhibitory activity was greater than 60%, and many spores germinated when the activity was less than 60% (Figure 1). Among the 342 fungal isolates, culture extracts from 89 isolates showed spore germination inhibitory activity of more than 80%, and 20 isolates showed activity of more than 90% (Appendix A). By classification of entomopathogenic fungi, approximately 80% or more of the germination inhibitory activity against *Nosema* was observed in 15 species of 10 genera among 24 species from 18 genera (Table 1). A total of 50 of the 126 isolates of *Beauveria* spp., 18 of the 81 isolates of *Metarhizium* spp., 11 of the 40 isolates of *Cordyceps* spp., 3 of the 17 isolates of *Pochonia* spp., and 2 of the 12 isolates of *Paecilomyces* spp. showed germination inhibitory activity on *Nosema* spores. As various activities were confirmed in various fungi, culture extracts of 89 isolates showing spore germination inhibitory activity of 80% or more were prepared again by quantitative inoculation of fungi, and the germination inhibitory activity on *Nosema* spores was re-evaluated.

### 3.2. Inhibitory Activity of Fungal Culture Extracts on Nosema Spore Germination

For 89 selected entomopathogenic fungi, culture extracts were prepared by inoculation with the same conidia concentration, then the extracts were diluted in distilled water at concentrations of 100, 10, and 1% to evaluate the germination inhibitory activity on *Nosema* spores (Figure 2). As a result, 44 of the 89 fungal culture extracts showed a high spore germination inhibitory activity of 80% or more, even at a diluted concentration of 1% and the extract stock solution. In particular, the decrease in activity according to the concentration of the extract of these isolates did not exceed 20% of the activity difference between the original stock solution and the diluted concentration of 1%. The activity of the remaining 45 isolates decreased significantly depending on the concentration of the extract, showing activity from approximately 50% to 5% at 1% extract concentration. A total of 25 isolates of *Beauveria* spp., 6 isolates of *Met. anisopliae*, 6 isolates of *Cordyceps* spp., 2 isolates of *Paecilomyces* spp., 2 isolates of *Pochonia* spp., and 3 other fungal isolates showed high activity similar to the stock solution even in the 100-fold diluted extract (Table 2). The following experiment was conducted using these 44 fungal culture extracts showing high spore germination inhibitory activity even at 100-fold dilution concentrations.

### 3.3. Mechanism of Inhibitory Activity of Fungal Culture Extracts on Nosema Spore Germination

To investigate the mechanism of inhibitory activity of 44 fungal culture extracts on *Nosema* spore germination, each fungal extract was treated on *Nosema* spores for 2 h, and after removal of the treated extracts, spore germination was observed. As a result, the fungal isolates showing a germination inhibition rate of approximately 60% or more even after removal of the culture extract were *M. anisopliae* 296, *Pae. marquandii* 364, *Poc. bulbillosa* 60, *Bea. brongniartii* 183, and *Bea. bassiana* 35, 161, and 59 (Figure 3). Among the remaining other fungal isolates, five isolates showed a germination inhibition rate of approximately 30% or more even after removing the extract, and the extracts of the other isolates showed no anti-germination activity or very low inhibition activity of approximately 5% or 20%. If the fungal culture extract showed spore germination inhibitory activity even after removal, it was determined that *Nosema* spores were inactivated by the culture extract, and the *Nosema* spore germination inhibitory activity of the culture extract was judged to be fungicidal activity. The inhibitory effect of fungal extracts on honeybee nosemosis was continuously evaluated using the culture extracts of 6 fungal isolates with 60% or more fungicidal activity, except for *Bea. bassiana* 161, which showed the lowest fungicidal activity.

### 3.4. Influence of Fungal Culture Extracts on Honeybees

Prior to the evaluation of the inhibitory activity of the culture extract on *Nosema* infection in honeybees, the influence of the culture extracts on the lifespan of honeybees were evaluated. Feeding with acetone, used as a solvent for the preparation of culture extract, showed a similar survival rate as that of the untreated honeybee group at concentrations of up to 2%. However, at a concentration of 4%, the survival rate of honeybees decreased after 4 days of treatment, and the final survival rate showed a difference of approximately 5% from the untreated group (Appendix A). When evaluating the nosemosis inhibitory activity of the fungal culture extract on honeybees, the concentration of acetone actually used does not exceed 0.2%. Therefore, these results indicate that acetone has little effect on the lifespan of honeybees. The influence of fungal culture extracts on honeybees was evaluated at concentrations of 1% and 10%. As a result of treatment with 1% diluted culture extract, 5 fungal isolates except *Bea. brongniartii* 183 had no significant influence on the survival rate of honeybees (Figure 4A). Even at 10% dilution, only *Bea. brongniartii* 183 partially reduced the survival rate of honeybees, so it was excluded from further experiments (Figure 4B).

### 3.5. Inhibitory Effect of Culture Extract on Honeybee Nosemosis

To evaluate the effect of inhibiting honeybee nosemosis by fungal culture extracts, honeybees were treated with each culture extract at a concentration of 5%, and the survival rate of honeybees and *Nosema* spore production were evaluated. The nosemosis inhibitory effect of the culture extract was evaluated in two ways: honeybees treated with the extract before *Nosema* infection and after *Nosema* infection. After treating each culture extract before infection with *Nosema*, *Pae. marquandii* 364 culture extract only improved the survival rate of honeybees by approximately 13% compared to *Nosema*-infected honeybees (Figure 5A). Other fungal culture extracts showed similar or lower survival rates of honeybees compared to *Nosema*-infected honeybees. Treatment of culture extracts of *M. anisopliae* 296, *Pae. marquandii* 364, *Bea. bassiana* 59, and *Poc. bulbillosa* 60 reduced spore production by approximately 65%, 32%, 80%, and 33%, respectively, compared to that of honeybees infected with *Nosema* alone (Figure 6A).

When honeybees were infected with *Nosema,* then treated with culture extracts, the extracts of *Poc. bulbillosa* 60 and *Pae. marquandii* 364 showed approximately 11% and 8% increases in honeybee survival rates, respectively, while the survival rates of the others decreased slightly comparing to *Nosema*-infected honeybees (Figure 5B). *Nosema* spore production was reduced compared to *Nosema*-infected honeybees in all fungal culture extract treatments (Figure 6B). These results suggested that the culture extracts of *Pae. marquandii* 364 and *Poc. bulbillosa* 60 were effective in inhibiting honeybee nosemosis.

## 4. Discussion

This study was conducted to screen and select culture extracts of entomopathogenic fungi that are effective for the control of honeybee nosemosis. Among the 342 entomopathogenic fungal isolates of 24 species from 18 genera tested, the inhibitory effect of fungal culture extracts on *Nosema* spore germination was shown in more than 95% of the isolates (Table 1 and Appendix A). These results suggested that the germination inhibitory substances on *Nosema* spores were contained in the fungal culture extract and were consistent with previous studies showing that entomopathogenic fungal metabolites had various biological activities. Reportedly, entomopathogenic fungal metabolites have antimicrobial activity against various bacteria and fungi, so sufficient antifungal activity could be expected for unicellular fungal parasites, such as *Nosema,* in our study [42,43,44,45]. To our knowledge, this is the first report showing that culture extracts of various diverse types of fungi have anti-*Nosema* activity.

To determine the most effective culture extract among various fungal culture extracts with anti-*Nosema* activity, six fungal culture extracts showing high anti-germination effects at a 100-fold diluted concentration and even when removed after treatment with the extract were selected and used for further testing anti-*Nosema* activity (Figure 2 and Figure 3). In our study, the germination inhibitory activity of the culture extract on *Nosema* spores was evaluated by dividing the fungistatic activity, which is inhibited only when the extract is present, and the fungicidal activity, which inactivates the spores by the extract [52]. It was judged that the fungicidal active culture extract was more effective in controlling nosemosis, and such fungal isolates were selected. In addition, the nosemosis inhibitory activity of the culture extract was shown as a preventive effect in one fungal isolate (Figure 5A) and a control effect in two fungal isolates (Figure 5B). However, the survival rate of honeybees in the experimental groups treated with the culture extract was lower than that of the control group not treated with the culture extract. These results are presumed to be due to the toxicity of the culture extract against honeybees. Although the honeybee survival rate was not significantly reduced compared to that in control under treatment with 10% of each culture extract, it is presumed that the low toxicity of the culture extract increased the susceptibility of honeybees to *Nosema*, thereby lowering the survival rate. However, it was confirmed that the production of *Nosema* spores was reduced by treatment with the culture extract (Figure 6). This result suggested that all the culture extracts could inhibit the proliferation of *Nosema*. As a particularly noteworthy result, treatment with the culture extracts of *Poc. bulbillosa* 60 and *Pae. marquandii* 364 showed a higher honeybee survival rate than uninfected honeybees, confirming that this culture extracts not only inhibit nosemosis but also increase the lifespan of honeybees. However, it has been shown that these effects may vary depending on the concentration of these culture extracts. When only the culture extract was treated at a 10% concentration, the survival rate of honeybees was not higher than that of the control group, but at a 1% concentration, the culture extracts of *Poc. bulbillosa* 60 and *Bea. bassiana* 59 partially increased the survival rate of honeybees (Figure 4). To date, there has been no report on substances showing positive activity on the lifespan of honeybees among entomopathogenic fungal culture extracts. Further research should identify the metabolites that showed positive effects in this work. Additionally, it should be determined whether the same metabolites have anti-*Nosema* activity and a positive effect on the lifespan of honeybees.

Various entomopathogenic fungal metabolites have been reported, and the most representative substance is beauvericin from *Bea. bassiana* and destruxins from *Metarhizium* spp., which are the most actively studied and utilized [43,53,54,55,56]. These substances have both antifungal activity and insecticidal activity. Other substances known to have antifungal activity include muscodorin, oosporein, patulin, enniatins, pradimicin, flavoglaucin, and terpenoids [43,45,55,57,58]. However, among these substances, inhibitory activity against *Nosema* has not been reported. Furthermore, since various fungal metabolites show various antifungal activities, the possibility that anti-*Nosema* active substances exist among fungal metabolites is considered sufficient. In previous reports on *Poc. bulbillosa* and *Pae. marquandii*, which showed the highest *Nosema* inhibitory activity in our study, we could not find any *Nosema* inhibitory activity. However, the antimicrobial activity and insecticidal activity of the metabolites in *Poc. chlamydosporia*, *Pae. variotii*, and *Pae. lilacinus* have been reported [59,60,61]. Through further research on the metabolites of *Poc. bulbillosa* 60 and *Pae. marquandii* 364, it may be possible to develop an effective control agent for honeybee nosemosis.

## 5. Conclusions

Culture extracts of 342 entomopathogenic fungal isolates of 24 species from 18 genera were used to evaluate the germination inhibitory activity on *Nosema* spores to search for fungal culture extracts with effective inhibitory activity against *Nosema*. As a result, inhibitory activity was observed in all fungal culture extracts except for 2 genera and 1 species of fungus, and high inhibitory activity of approximately 80% or more was shown in fungal isolates of 15 species and 10 genera. Among them, the culture extracts of *Poc. bulbillosa* 60 and *Pae. marquandii* 364, which have fungicidal activity at low concentrations, not only effectively inhibited honeybee nosemosis but also prolonged the lifespan of honeybees.

## Figures and Tables

**Figure 1 insects-14-00538-f001:**
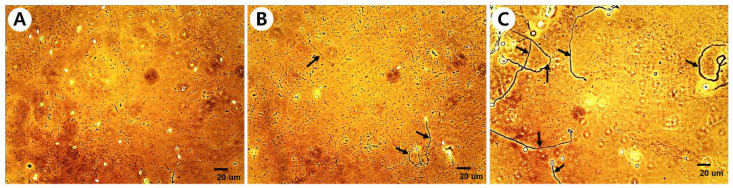
Spore germination inhibitory activity of fungal culture extract on *Nosema ceranae*. (**A**) more than 80% inhibition; (**B**) more than 60% inhibition; (**C**) less than 60% inhibition. The scale bar is 20 μm. Arrows indicate mycelia that have grown after germination.

**Figure 2 insects-14-00538-f002:**
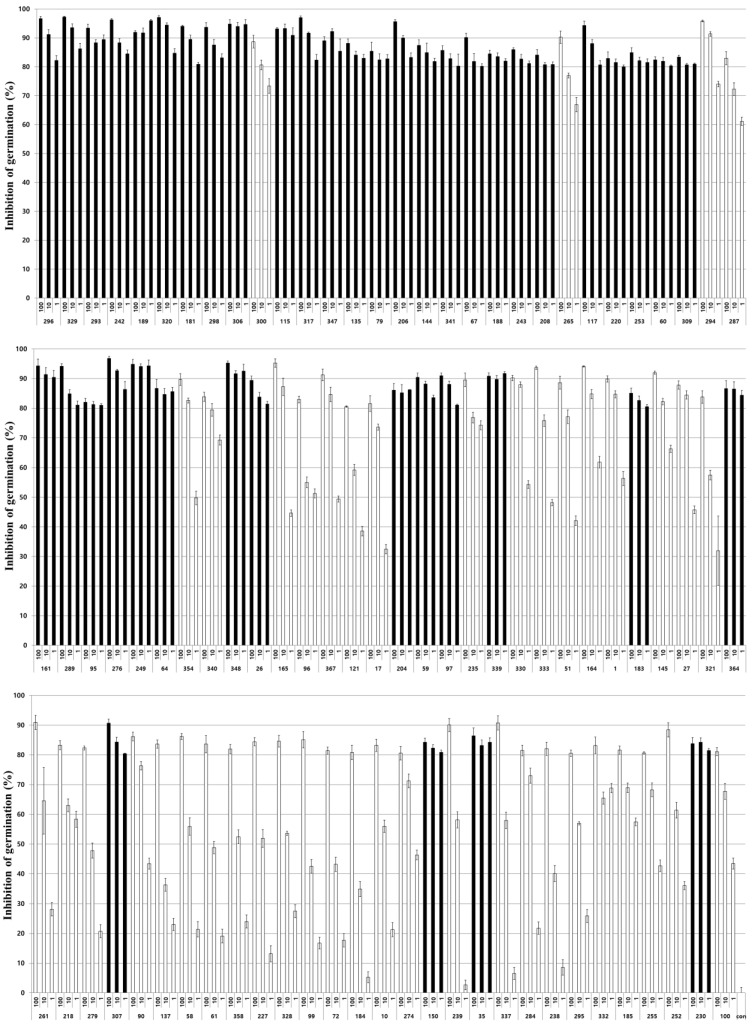
Inhibition rate of *Nosema ceranae* spore germination in response to 89 entomopathogenic fungal culture extracts. Each extract was used to treat *Nosema ceranae* spores at concentrations of 100%, 10%, and 1%, then the germination rate was determined. Fungal isolates exhibiting an inhibition rate of 80% or more are indicated by black bars. Data show the mean ± SE.

**Figure 3 insects-14-00538-f003:**
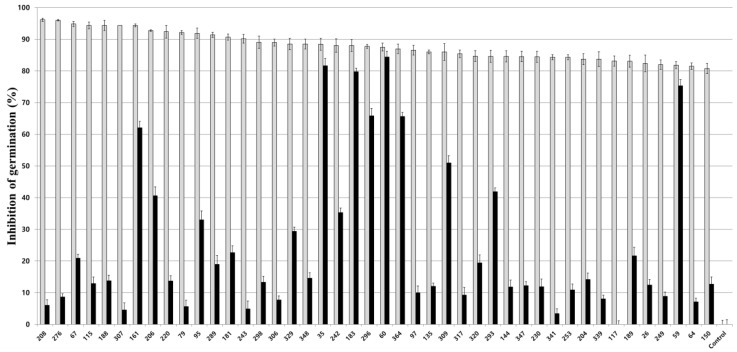
Evaluation of the fungistatic and fungicidal activities of entomopathogenic fungal culture extracts on *Nosema ceranae* spores. After treatment with the fungal culture extract on *Nosema ceranae* spores, germination was observed without removing the extract (grey bar) or after removing the extract (black bar). Data show the mean ± SE.

**Figure 4 insects-14-00538-f004:**
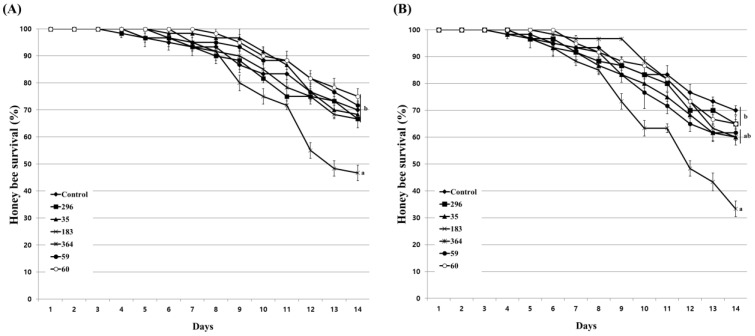
Honeybee survival under 1% (**A**) and 10% (**B**) fungal culture extract treatments. A mixture of fungal culture extract and 50% sucrose solution was fed to honeybees. Afterward, the survival rate of honeybees was determined for 14 days. The control group was fed only a 50% sucrose solution. Data show the mean ± SE. Values with different letters are significantly different (*p* < 0.05, SNK test in one-way ANOVA) at 14 days.

**Figure 5 insects-14-00538-f005:**
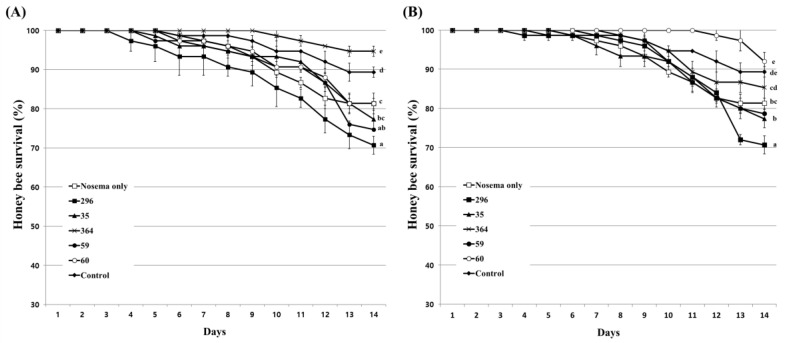
Survival rate of honeybees against *Nosema ceranae* infection. Honeybees were treated with 5% fungal culture extract before (**A**) and after (**B**) *Nosema* infection. The control group and Nosema-only group were fed a 50% sucrose solution and a mixture containing *Nosema* spores, respectively. Data show the mean ± SE. Values with different letters are significantly different (*p* < 0.05, SNK test in one-way ANOVA) at 14 days.

**Figure 6 insects-14-00538-f006:**
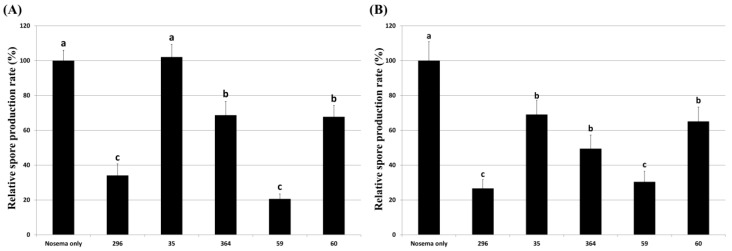
*Nosema* spore production in *Nosema*-infected honeybees treated with 5% fungal culture extract before (**A**) or after (**B**) *Nosema ceranae* infection. The relative ratio of the spore production of each treatment group to the spore production of honeybees infected with Nosema only was shown. Data show the mean ± SE. Values with different letters are significantly different (*p* < 0.05, SNK test in one-way ANOVA).

**Table 1 insects-14-00538-t001:** Entomopathogenic fungi showing more than 80% *Nosema ceranae* germination inhibition activity.

Fungus	No. of Tested Isolates	No. of Isolates Showing the Inhibition of Spore Germination
All fungal isolates	342	89 (26%) *
*Acremonium strictum*	1	0 (0%)
*Aspergillus lentulus.*	5	0 (0%)
*Aspergillus versicolor*	3	1 (33.3%)
*Beauveria bassiana*	110	48 (43.6%)
*Beauveria brongniartii*	8	1 (12.5%)
*Beauveria pseudobassiana*	8	1 (12.5%)
*Bionectria ochroleuca*	7	0 (0%)
*Clonostachys rosea*	1	0 (0%)
*Cordyceps farinosa*	12	5 (41.7%)
*Cordyceps fumosorosea*	6	1 (16.7%)
*Cordyceps javanica*	22	5 (22.7%)
*Fusarium oxysporum*	2	0 (0%)
*Lecanicillium* spp.	8	1 (12.5%)
*Metarhizium anisopliae*	64	15 (23.4%)
*Metarhizium lepidiotae*	1	0 (0%)
*Metarhizium pemphigus*	16	3 (18.8%)
*Mucoromycotina* spp.	1	0 (0%)
*Myrothecium* spp.	5	0 (0%)
*Paecilomyces lilacinus*	7	1 (14.3%)
*Paecilomyces marquandii*	5	1 (20%)
*Paraconiothyrium sporulosum*	2	1 (50%)
*Phialocephala* spp.	1	0 (0%)
*Pochonia bulbillosa*	16	3 (18.8%)
*Pochonia rubescens*	1	0 (0%)
*Simplicillium aogashimaense*	1	1 (100%)
*Simplicillium* sp.	2	0 (0%)
*Tolypocladium album*	23	1 (4.3%)
*Tolypocladium cylindrosporum*	3	0 (0%)
*Verticillium insectorum*	1	0 (0%)

* Relative % to the number of isolates (species or genus).

**Table 2 insects-14-00538-t002:** Entomopathogenic fungi showing the inhibition of *Nosema ceranae* spore germination by more than 80% in 1% concentration culture extract.

Fungus	No. of Isolate (%)
*Aspergillus versicolor*	1 (2.3) *
*Beauveria bassiana*	24 (54.5)
*Beauveria brongniartii*	1 (2.3)
*Cordyceps farinosa*	4 (9.1)
*Cordyceps fumosorosea*	1 (2.3)
*Cordyceps javanica*	1 (2.3)
*Lecanicillium* spp.	1 (2.3)
*Metarhizium anisopliae*	6 (13.6)
*Paecilomyces lilacinus*	1 (2.3)
*Paecilomyces marquandii*	1 (2.3)
*Pochonia bulbillosa*	2 (4.5)
*Tolypocladium album*	1 (2.3)

* Relative % to number of isolates (species or genus).

## Data Availability

The data are available at reasonable request from the corresponding author.

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
