# Peer review of "Screening of Entomopathogenic Fungal Culture Extracts with Honeybee Nosemosis Inhibitory Activity"

_insects, 2023, doi:10.3390/insects14060538_

Round 1

Reviewer 1 Report

The research is original and interesting, and the results can be applicable because they encourage further work that can lead to the formulation of nosema control preparations based on extracts of entomopathogenic fungi.

However, the paper is written in poor English, and requires correction by a professional English proofreader.

When reading, one gets the impression that different parts were written by different authors, so there are easy-to-understand and hard-to-understand parts.

While reading the paper, I repeatedly wondered and checked whether the effect of METABOLITES of entomopathogenic fungi or culture extracts from entomopathogenic fungi was tested in the paper. No metabolites were identified (or listed) in sections MATERIAL & METHODS and RESULTS (not even in Tables and Figures and their legends). So, only fungal culture extracts were examined in the paper, and no metabolites. Thus, I strongly suggest authors to remove the word „metabolite(s)“ from the TITLE, SIMPLE SUMMARY, ABSTRACT, KEY WORDS and CONCLUSIONS. Also, in Line 84 where the purpose of the study is defined, the word „metabolites“ should be removed.

The SIMPLE SUMMARY needs to be rewritten because it has too much detail and does not clearly show the essence of the research.

Line 52 – after ’human-related factor’ I suggest that you refer to Stanimirovic et al. 2019 (reference below)

·        Stanimirovic Z, Glavinic U, Ristanic M, Aleksic N, Jovanovic N, Vejnovic B, Stevanovic J: Looking for the causes of and solutions to the issue of honey bee colony losses. Acta Veterinaria-Beograd 2019, 69(1): 1–31.

Line 59 – The negative effects of Nosema infections do not apply to all geographic areas, so I suggest you add that.

Line 62 – after „reduced immune function“ I suggest that you refer to: Glavinic et al. 2017, 2021a, and 2021b. (references below)

·        Glavinic, U.; Stevanovic, J.; Ristanic, M.; Rajkovic, M.; Davitkov, D.; Lakic, N.; Stanimirovic, Z. Potential of fumagillin and Agaricus blazei mushroom extract to reduce Nosema ceranae in honey bees. Insects 2021, 12, 282.

·        Glavinic, U.; Rajkovic, M.; Vunduk, J.; Vejnovic, B.; Stevanovic, J.; Milenkovic, I.; Stanimirovic, Z. Effects of Agaricus bisporus mushroom extract on honey bees infected with Nosema ceranaeInsects 2021, 12, 915.

Line 72 – next to the ’plant extracts’ I suggest you to add ’Agaricus spp. mushroom extracts“ and refer to: Glavinic et al. 2021a and 2021b. (references below)

·        Glavinic, U.; Stevanovic, J.; Ristanic, M.; Rajkovic, M.; Davitkov, D.; Lakic, N.; Stanimirovic, Z. Potential of fumagillin and Agaricus blazei mushroom extract to reduce Nosema ceranae in honey bees. Insects 2021, 12, 282.

·        Glavinic, U.; Rajkovic, M.; Vunduk, J.; Vejnovic, B.; Stevanovic, J.; Milenkovic, I.; Stanimirovic, Z. Effects of Agaricus bisporus mushroom extract on honey bees infected with Nosema ceranaeInsects 2021, 12, 915

Line 88, Line 146, Line 147 (check whole manuscript - you wrote ’50% sucrose’. Did you mean ’50% (w/vsucrose solution

Line 89-90 – ’Nosema ceranae spores were provided’. I am wondering what you exactly obtained from the Sericulture and Apiculture Division .... Nosema spores or Nosema-infected bees? In subsection 2.3. (Lines 116-132) you described in detail how you produced infective spores.

Line 93 and Table S1: What is the difference between isolates of the same fungal species?  

Line 122-123 : ’The homogenate increased the volume up to 1 mL’. I do not understant this.

Line 147-148 – This is unclear: ’containing the same concentration of acetone as the treated solvent was fed once’.

Line 151:  (Subsection 2.6.) The subtitle is incorrecly written! I suppose you wanted to say: ’Inhibitory activity of fungal culture extracts on Nosema infection in honey bees’.

Line 177-178 – You wrote ’All fungal isolates used in the experiment were mostly only expressed with numbers for convenience’. This sentence is not necessary here.  It is much better to put it in the Legend of Tabele S2.

Line 203 (Subsection 3.2.) – The subtitle is incorrecly written! I suppose you wanted to say: ’Inhibitory activity of fungal culture extracts on Nosema spore germination’.

Line 248 (Subsection 3.3.) –  Again, unclear subtitle!  

Line 250 – unclear.

Line 259 – 261 – unclear.

Line 261 – unclear.

Line 271-285 – unclear.

Line 296-297 – unclear.

Lines 172-174 and Lines 274-280 – check the effects of acetone.

Line 323-325 – The title of Figure 5 – unclear.

A lot of things are repeated in the DISCUSSION, and the comparison with the results of other researches should prevail (for example, the whole part in lines 344-375 lacks references). Please try to combine and compare your results with those of previous studies.

Line 332, 339,  – Which metabolites?  You did not identify (specify) any of them. I suggest you to write ’fungal culture extracts“.

Line 341, 342 ... – replace ’fungi such as Nosema’ with ’unicellular fungal parasite such as Nosema’.

Line 347 – correct the sentence as follows: ’selected and used for testing anti-Nosema activity’.

Lines 349-350 – I am wondering if the division into fungistatic and fungicidal is defined here for the first time? If not, please refer to literature sources.  

Line 352-354 – unclear.

Lines 356-357 – unclear.

357-358 – In relation to the toxicity, is there any reference you could refer to?

Line 359-361 – good assumption, but do you have a reference for comparison?

Line 371 ... you should compare this with the findings of other studies.    

Line 373 – Please delete ’In our results’... I suggest to replace the whole sentence (Line 373-375) with this one: ’Further research should identify the metabolites that showed positive effects in this work. Also, it should be determined whether the same metabolites are those that have anti-Nosema activity and those that have a positive effect on the lifespan of bees.’

Line 384 – ’in previous report’ – please specify the report.

Line 388-390 – unclear.

Lines 395-398 – metabolites ... metabolites ... As I have already explained, this term is not adequate!

Line 398 - cultured metabolites!?!? please be careful!

Line 402 – What is with lifespan of honey bees ... I think  the effect on lifespan should be mentioned here.

Very bad English, so the whole manuscript has to be corrected by a professional English proofreader.

Reviewer 2 Report

The article submitted for review concerns the possibility of inhibition of nosemosis in bees by metabolites of entomopathogenic fungi. This is an extremely important issue because of the role of the honeybee in the human economy and the functioning of natural ecosystems. The authors examined the metabolites of 342 fungal isolates (18 genera, 24 species). The results of the study indicate that metabolites of 2 fungal species (Pochonia bulbillosa and Paecilomyces marquandii) may find application in nosemosis control programs.

Overall, the manuscript is well written and can be accepted for publication. Minor comments:

L281: correct fung6al to fungal

Throughout the manuscript, there are no references to the results of statistical tests. Complete.

Only one of the graphs indicates statistically significant differences between groups. Complete.

Round 2

Reviewer 1 Report

I read the revised version and I can conclude that the authors tried their best to correct the paper, but it is clear that it was not reviewed by a professional proofreader or a person who is proficient in English. Therefore, the paper cannot be accepted until it is written clearly and correctly. My second complaint is related to the lack of answers to the questions I asked. The general rule is that authors send responses to each reviewer's comment. Did the authors forget to send their responses? Since they adopted most of my suggestions and made changes based on them, I list the issues they ignored and and I ask the authors to answer them

I list the questions they ignored and I ask the authors to answer them:

Line 89-90 – ’Nosema ceranae spores were provided’. I am wondering what you exactly obtained from the Sericulture and Apiculture Division .... Nosema spores or Nosema-infected bees? In subsection 2.3. (Lines 116-132) you described in detail how you produced infective spores.

Line 92-93 and Table S1: What is the difference between isolates of the same fungal species?  

Line 122: ’The homogenate increased the volume up to 1 mL’. I do not understant this.

Line 250 – unclear. Now Lines 247-248:  Still unclear “the extracts of each fungal isolate were treated on Nosema spores 247 for 2 h, and after removal of the treated extracts, ……”

Last time I said that the paper is written in poor English, and requires correction by a professional English proofreader. However, the authors did not ask a professional proofreader or a person who is proficient in English, so the English is still poor. 

Round 3

Reviewer 1 Report

Lines 90-91: Sorry, but the inserted sentence does not contain the answer to my question; that sentence is redundant, please delete it. Since you did not write how you ensured viability of Nosema ceranae spores and you also did not check the viability before inoculation, I think you should have taken highly infected bees rather than “purified N. ceranae spores” from another institution (because N. ceranae spores rapidly lose viability over time in the refrigerator and almost completely lose after freezing (Fenoy et al., 2009; Fries, 2010; reviewed in: Fries et al. (2013) Standard methods for Nosema research. Journal of apicultural research. 2013 Jan 1;52(1):1-28. - https://www.tandfonline.com/doi/abs/10.3896/ibra.1.52.1.14.

Lines 131-133: I found a contradiction here: “spore pellet was resuspended in sterile distilled water and stored at room temperature until use” …. “and the spore suspension was freshly prepared before use”. If you store spore suspension until use, that might mean that you waited a long time, … consequently, that means that you did not use fresh suspension”. I suggest you: 1) to replace the part “spore suspension was freshly prepared before use” at the beginning of subsection, and 2) to delete part “stored at room temperature until use”.

Figure S1. The first sentence has no meaning “Germination of Nosema ceranae spores by acetone solvent treatment”. Please correct English.

Lines 275-282: Seriously syntax errors. English correction is obligatory.

Still bad, Extensive editing of English language required

Author Response

3rd-response to Reviewer 1 Comments

Thank you for your comments.

We have read and considered your comments carefully, and revised the manuscript accordingly.

All the concerns have been addressed in the 3rd revised manuscript.

Point 1: Lines 90-91: Sorry, but the inserted sentence does not contain the answer to my question; that sentence is redundant, please delete it. Since you did not write how you ensured viability of Nosema ceranae spores and you also did not check the viability before inoculation, I think you should have taken highly infected bees rather than “purified N. ceranae spores” from another institution (because N. ceranae spores rapidly lose viability over time in the refrigerator and almost completely lose after freezing (Fenoy et al., 2009; Fries, 2010; reviewed in: Fries et al. (2013) Standard methods for Nosema research. Journal of apicultural research. 2013 Jan 1;52(1):1-28. - https://www.tandfonline.com/doi/abs/10.3896/ibra.1.52.1.14.

Response 1:  Thank you for your comments. We modified it according to your comments.

Point 2: Lines 131-133: I found a contradiction here: “spore pellet was resuspended in sterile distilled water and stored at room temperature until use” …. “and the spore suspension was freshly prepared before use”. If you store spore suspension until use, that might mean that you waited a long time, … consequently, that means that you did not use fresh suspension”. I suggest you: 1) to replace the part “spore suspension was freshly prepared before use” at the beginning of subsection, and 2) to delete part “stored at room temperature until use”..

Response 2: We modified it according to your comments and added viability related content.

Point 3: Figure S1. The first sentence has no meaning “Germination of Nosema ceranae spores by acetone solvent treatment”. Please correct English.

Response 3: We modified it according to your comments.

Point 4: Lines 275-282: Seriously syntax errors. English correction is obligatory

Response 4: We modified it according to your comments.

Point 5: Still bad, Extensive editing of English language required.

Response 5: We have once again corrected the English sentences throughout the manuscript.